# Multi-Theory Model and Predictors of Likelihood of Accepting the Series of HPV Vaccination: A Cross-Sectional Study among Ghanaian Adolescents

**DOI:** 10.3390/ijerph17020571

**Published:** 2020-01-16

**Authors:** Matthew Asare, Peter Agyei-Baffour, Beth A. Lanning, Alex Barimah Owusu, Mary E. Commeh, Kathileen Boozer, Adofo Koranteng, Lori A. Spies, Jane R. Montealegre, Electra D. Paskett

**Affiliations:** 1Department of Public Health, Robbins College of Health and Human Sciences Baylor University, One Bear Place, Waco, TX 97343, USA; Beth_Lanning@baylor.edu; 2School of Public Health, Kwame Nkrumah University of Science and Technology, Kumasi, Ghana; agyeibaffourp@gmail.com (P.A.-B.); adofokoranteng@gmail.com (A.K.); 3Department of Geography and Resource Development, University of Ghana, Legon, Ghana; owusuba@yahoo.com; 4Ghana Health Services, Non-Communicable Disease Control, Accra, Ghana; commehefe@gmail.com; 5Louise Herrington School of Nursing, Baylor University, Dallas, TX 75246, USA; kathileen_boozer@baylor.edu (K.B.); lori_spies@baylor.edu (L.A.S.); 6Department of Pediatrics and Dan L Duncan Comprehensive Cancer Center, Baylor College of Medicine, Houston, TX 77030, USA; jrmontea@bcm.edu; 7Department of Internal Medicine, Division of Cancer Prevention and Control in the College of Medicine, College of Medicine, The Ohio State University, Columbus, OH 43210, USA; electra.paskett@osumc.edu

**Keywords:** adolescents, HPV vaccination, multi-theory model

## Abstract

HPV vaccines are efficacious in preventing HPV related cancers. However, the vaccination uptake in Ghana is very low. Studies that utilize theoretical frameworks to identify contributory factors to HPV vaccination uptake in Ghana are understudied. We used multi-theory model (MTM) constructs to predict initiation and completion of HPV vaccination series in Ghanaian adolescents. Adolescents (*n* = 285) between the ages of 12 and 17 years old were recruited from four selected schools in Ghana to participate in the cross-sectional study. Linear regressions were used to analyze the data. Most participants were female (91.2%) and senior high school students (60.0%). Many of the participants had neither heard about HPV (92.3%) nor HPV vaccinations (95.4%). Significant predictors of adolescents’ likelihood of getting the first dose of HPV vaccination were perceived beliefs and change in a physical environment (*p* < 0.001), with each variable accounting for 6.1%and 8.8% of the variance respectively. Significant predictors of adolescents’ likelihood of completing HPV vaccination recommended series were perceived beliefs, practice for change, and emotional transformation (*p* < 0.001), with each variable accounting for 7.8%, 8.1%, and 1.1% of the variance respectively. Findings underscore important opportunities for developing educational interventions for adolescents in Ghana to increase the HPV vaccination uptake.

## 1. Introduction

African governments, non-governmental, and philanthropic organizations have concentrated much of their efforts on preventing malaria and HIV/AIDS in the African continent. However, less attention has been given to Human Papillomavirus (HPV) related cancers, which are silently claiming the lives of Africans in great numbers. HPV is the most common sexually transmitted diseases in the world [1,2]. HPV infection has a long incubation period with symptoms often occurring years after the initial infection. The latent onset of symptoms is problematic because exposed individuals may be unaware of their disease status yet have the potential to spread the virus to another individual. Depending on the integrity of the person’s immune system, the virus may be self-eradicated without the person experiencing further health problems [3]. However, persistent infection with oncogenic HPV causes nearly all cervical cancers and many vulvar, vaginal, penile, anal, oral, oropharyngeal, neck, and head cancers [4]. The World Health Organization (WHO) reported that there were approximately 528,000 new cases of HPV and 266,000 HPV related deaths worldwide in 2012 [5]. In 2018, approximately 190,000 new HPV related cancer cases were diagnosed, (representing 17.97% of cancer cases) and over 135,200 HPV related cancer deaths (representing 19.50% of cancer deaths) occurred in Africa [6]. In 2018 in Ghana, over 3300 new cases of HPV related cancers were diagnosed (representing 14.47% of cancer cases) and 2977 HPV cancer deaths (representing 19.73% of cancer deaths) were reported [6]. The most common HPV cancer type in Ghana is cervical, which is the leading cause of female cancer deaths in the country [7,8,9]. It is estimated that over nine million women aged 15 years and above in Ghana are at risk for cervical cancer [7]. These HPV caused cancers and the associated morbidity and mortality are preventable.

Fortunately, prophylactic HPV vaccines such as Cervarix^®^, Gardasil^®^, and Gardasil 9^®^ are efficacious in preventing HPV related cancers [10,11,12,13]. Cervarix^®^ is a bivalent vaccine that targets HPV 16 and 18, Gardasil^®^ is a quadrivalent vaccine that additionally targets HPV 6 and 11 which cause genital warts. Gardasil 9^®^ is a nonavalent vaccine that has been licensed in the United States of America (USA), Europe, and other high-income countries and targets additional oncogenic HPV serotypes: 31, 33, 45, 52, and 58. Studies show that the HPV vaccines produce a higher immune response in 9–14 year-olds than they do in 15–26 year-olds, and as such to increase efficacy the vaccination is recommended in the early years of adolescence [14]. The World Health Organization (WHO) through its Strategic Advisory Group of Experts on Immunization (SAGE) currently recommends two doses of vaccination for girls aged 9 to 14 years [10,15,16,17,18]. However, the Advisory Committee on Immunization Practices (ACIP) a three-dose ‘catch-up’ vaccination is recommended for females aged 15 through 26 years and males aged 15 through 21 years who have not been previously vaccinated [14]. In 2018, the United States Food and Drug Administration (FDA) expanded HPV vaccination recommendations to include both men and women between the ages of 27 and 45 years [19]. Since the introduction of the HPV vaccines, there has been a significant decrease in the prevalence of HPV infection and its subsequent diseases in the US, Australia, and other parts of the world [20,21]. Low-and middle-income countries, including Ghana, have the lowest global vaccination rates [7]. In Ghana, the HPV vaccination was first introduced in 2013 [22] and currently, both Cervarix^®^ and Gardasil^®^, are sold in Ghana. Since 2013, there have been two HPV vaccination demonstration projects completed in Ghana, one by Global Alliance for Vaccines and Immunizations (GAVI) [23,24] and the other one through GARDASIL^®^ Access Program (GAP) [24]. Unfortunately, those HPV vaccination programs were short-lived and GAVI has since concentrated their efforts on other competing health priorities in Ghana [23]. A complicating factor is the lack of data on current vaccine uptake in Ghana pointing to the need to understand factors that contribute to low vaccination rates [25].

Barriers to HPV vaccination in sub-Saharan Africa include structural factors such as inadequate infrastructure, limited trained health workers, vaccine cost, stigma, and lack of knowledge [26,27,28]. Furthermore, studies specifically identifying challenges to and facilitators of HPV vaccinations in Ghana are relatively rare. While Edwin (2010) identified religious, political, socioeconomic, and ethical challenges to HPV vaccination in Ghana [9], studies that examined adolescents in middle school and high school awareness and knowledge about HPV vaccinations are limited. Identifying adolescents predisposing, enabling, reinforcing, and societal (cultural) factors that facilitate and hinder HPV vaccination is important for the development of effective interventions to increase the HPV vaccination behavior among adolescents in Ghana. Studies that utilize theoretical frameworks to identify contributory factors to HPV vaccination uptake in Ghana are limited. A novel multi-theory model of behavior change is a viable theoretical framework that can be used to predict both the initiation and completion of HPV vaccination behaviors in adolescents in Ghana.

### Theoretical Framework

The multi-theory-model (MTM) [29] is a behavior change theory with two main components: initiation of the behavior change, and continuation of the health behavior change. The initiation of behavior change includes three key constructs. First, is a participatory dialogue which is defined as the difference between advantages and disadvantages of engaging in a behavior (e.g., the initiation of HPV vaccination). Behavioral confidence is a second construct which is a function of individual certainty to perform a given health behavior change (e.g., confidence in getting HPV vaccination) in the future. Changes in the physical environment is a third construct that can be used to evaluate whether the availability and accessibility of physical factors (e.g., cost and availability of HPV vaccination) could facilitate a behavior change [30].

The continuation of a given behavior also involves three key constructs, emotional transformation, practice, and social environment. The emotional transformation construct is changing individuals’ emotions toward a given behavior change (e.g., making sure to complete the recommended doses of HPV vaccination). The construct of practice for change highlights the individuals’ deep consideration of a given behavior change and engaging in iterative processes to overcome barriers and remaining focused on sustaining a behavior change (e.g., keeping journals to remind one’s self about going for a second dose of HPV vaccination). The construct of changes in the social environment stipulates that creating social support from the environment (e.g., parental and primary care provider support for HPV vaccination) helps in sustaining a given health behavior [30]. Unlike other health behavioral theories that include only individual constructs of the MTM, including the combination of the constructs may result in more predictive power when operationalized in an intervention setting [30].

The inclusion of MTM in HPV vaccination research constitutes a new approach well supported in health promotion and public health research. The use of MTM has consistently shown to have high predictive power across multiple health-related behaviors [30]. Since its introduction, MTM has been used to predict physical activity behavior in adults [31], portion size behavior in adults, [32] and sleep behavior [33]. Furthermore, MTM includes constructs: (participatory dialogue [34], behavioral confidence [35,36], social environment, physical environment [35,37], and emotional transformation [38]) that have been empirically tested and proven to be efficacious in guiding intervention studies [29]. The purpose of this study was to identify factors that predict the likelihood of initiation and completion of HPV vaccination series in adolescents in Ghana using MTM constructs.

## 2. Materials and Methods

### 2.1. Sample Size Determination

A target sample size was calculated in the G*Power software, specifying an effect size of 0.10, an alpha of 0.05, power of 0.95, and 7 predictors. Based on the criteria, a total sample size of 226 was determined adequate for the study. To account for potential missing data the sample size was increased by 25% and therefore our recruitment target was 283 participants.

### 2.2. Recruitment Procedure

A convenience sample of adolescents was recruited from four schools in the Ashanti Region in Ghana to complete a 44-item survey based on MTM constructs. To be eligible, adolescents had to be 12 to 18 years and be able to read or understand English. The study protocol went through a series of approval processes before recruiting adolescents for the study. First, study approval was also obtained from the Ghana health services, Ghana education service, and school administrators. After obtaining these approvals, we worked with the school administrators to identify interested students and obtained parental permission before the child could assent to the study. All assenting adolescents were given a hard copy of the survey to complete and return to the researchers.

### 2.3. Measures and Reliability Assessment

We used a modified version of multi-theory model (MTM) questionnaire [30]. The reliability of the modified version of the survey was tested using test–retest reliability (correlation coefficient) and internal consistency (Cronbach alpha) methods. To assess the reliability of the tool, the questionnaire was administered twice among 40 adolescents at an interval of two weeks. The correlation coefficient and Cronbach alpha of 0.70 is considered acceptable [39]. The modified version consisted a 44-item survey assessing demographic characteristics, predictor, and outcome variables. Demographic variables included age, gender, education, religion, and evidence of health insurance. Two knowledge items, whether participants had heard about HPV and HPV vaccinations were included with a yes/no response option.

### 2.4. Predictors

The predictor variables were the constructs of MTM and the perceived beliefs subscale. The MTM constructs consisted of three subscales that predict the initiation of the HPV vaccination: participatory dialogue, behavioral confidence, and physical environment, and three subscales to predict the completion of the HPV vaccination recommended series: social environment, practice for change, and emotional transformation.

#### 2.4.1. Initiation Subscales Predictors

The participatory dialogue consists of the perceived benefits of HPV vaccination (5 items) and perceived barriers of HPV vaccination (5 items). The perceived benefits of HPV vaccination items included protection against cervical, head and neck, mouth and throat cancers and genital warts, improve the immune system, and feel better about overall health. The perceived barriers of the HPV vaccination items included side effects, the cost of the vaccine, inconvenience, pain, and possible effects on the reproductive system. To calculate the participatory dialogue score, the disadvantages scores were subtracted from the advantages scores. This scoring approach is consistent with the recommendation by theory’s author [30], The participatory dialogue score ranged from −25 to 25 where negative scores indicated less likely to participate and positive scores indicated more likely to participate. The behavioral confidence subscale (5 items) included the certainty of getting the HPV vaccination in the future without having insurance, with their present schedule, irrespective of the distance involved to get a vaccination, and vaccine side effects. The changes in physical environment subscale (3 items) included the certainty of arranging for payment, place, and transportation to get the first dose of vaccination. The composite scores were computed for the behavioral confidence and the changes in the physical environment items and the possible scores range were 1–25 and 1–15 respectively, where higher scores indicated the likelihood of getting the first dose of HPV vaccination.

#### 2.4.2. Completion of Recommended Dose Predictors

The changes in social environment subscale (3 items) includes certainty of getting support from a family member, friends, and health professionals for all the recommended series of HPV vaccination. The practice for change subscale (3 items) included certainty of being able to keep a record to monitor getting the recommended series, overcome barriers to getting recommended doses, and change schedule to be able to get the recommended doses. The emotional transformation subscale (3 items) included certainty related to being able to direct emotions, motivate oneself, and overcome self-doubt to getting the recommended doses. The composite scores were computed for the subscales and the possible score range was 1–15 where higher scores indicated the likelihood of completing the recommended doses of the vaccine.

#### 2.4.3. Additional Predictor

We also included a perceived belief variable as an additional predictor. The three perceived belief items assessed participants’ perceived belief about whether HPV can cause cancer, HPV vaccination can prevent HPV related cancer, and HPV vaccination was effective when adolescents are vaccinated in their pre-teen years. The possible score was 1–15 where higher scores indicated higher beliefs.

### 2.5. Outcome Variables

The two outcome variables for the study were the likelihood of adolescents getting the first dose of HPV vaccination within the next month and the likelihood of adolescents completing the recommended series of HPV vaccination with the next 12 months. The response options were somewhat likely, moderately likely, likely, very likely, and completely likely. The possible score range was 1–5 where 1 indicated less likelihood and 5 indicated a high likelihood of initiation or completion.

### 2.6. Statistical Analyses

Descriptive statistics were calculated on the demographic data to describe the sample. Univariate analysis was used to compare the mean difference of participants’ likelihood of getting the first dose and completing the recommended series of HPV vaccination. Hierarchical multiple regression analyses were conducted to predict HPV vaccination behaviors. All data were analyzed using the IBM Statistical Package for Social Sciences (SPSS) version 25.0.

### 2.7. Ethics Declarations 

The protocol was reviewed and approved by the Institutional Review Boards (IRBs) of Baylor University (IRB# 1355428) and Kwame Nkrumah University of Science and Technology in Ghana (IRB# CHRPE/AP /126/19).

## 3. Results

### 3.1. Sample Characteristics

The sample characteristics for adolescents are given in Table 1. A sample of 285 adolescents (M = 15.47, SD 1.80 years) participated in the study and provided evaluable data that were included in the analysis. Ten participants who did not provide data on the outcome variables were excluded from the final analysis. Most participants were female (91.2%), senior high school students (60.0%), Christians (96.8%), and did not have a primary care doctor or nurse (84.9%). The majority of the participants had not heard about HPV (92.3%) or HPV vaccination (95.4%).

### 3.2. Descriptive Statistics

Participant scores for change in a social environment (M = 11.79, SD = 2.81), change in the physical environment (M = 11.79, SD = 2.22), emotional transformation (M = 11.60, SD = 2.82) and practice for change (M = 11.55, SD = 2.86) were relatively high. However, participants’ participatory dialogue (i.e., advantage–disadvantages) of HPV vaccination (M = 2.98, SD = 5.30) was very low (Table 2).

### 3.3. Subscale Consistency and Test–Retest Reliability

The internal consistency for the subscales calculated using Cronbach alpha ranged from 0.78–0.96. The test–retest results for the subscales showed correlation coefficients (*r*) ranging from 0.65–0.92 (see Table 2).

### 3.4. Initiation of HPV Vaccination

Univariate analyses for initiation showed that participants’ sex and age were strongly associated with the likelihood of getting the first dose of HPV vaccination (*p* < 0.001). Female participants were more likely than males to report a willingness to get the first dose of the vaccine (M = 3.46 vs. M = 2.64; *p* < 0.01, respectively). Participants between ages 12 and 15 years were more likely than those between ages 16 and 18 years to report a willingness to get the recommended doses of the vaccination (mean difference 4.00 vs. 3.27; *p* < 0.01, respectively).

In hierarchical multiple regression analyses for initiation (see Table 3), the results indicated that 23.6% of the variance in adolescents’ likelihood of getting the first dose of HPV vaccination was explained by the model (R^2^ = 0.236, *F* (10, 274) = 8.488, *p* < 0.001). After controlling for the sample characteristics in the final model (see Model 3), the perceived beliefs and change in physical environment were the significant predictors of the adolescents likelihood of getting their first dose of HPV vaccination (*p* < 0.001), with each variable accounting for 6.1% and 8.8% of the variance respectively.

### 3.5. Completion of HPV Vaccination Series

Univariate analyses for completion showed that age, sex and education were strongly associated with the likelihood of getting the recommended doses of HPV vaccination (*p* < 0.001). Participants between ages 12 and 15 years were more likely than those between ages 16 and 18 years to report a willingness to get the recommended series of the vaccination (M = 4.02 vs. M = 3.26; *p* < 0.001, respectively). Females were more likely than males to report a willingness to complete the recommended doses of the vaccines (M = 3.79 vs. M = 1.64; *p* < 0.001, respectively).

In hierarchical multiple regression analyses (see Table 4), the results revealed that 36.0% of the variance in the likelihood of completing the recommended doses of HPV vaccination was explained the by the whole model (R^2^ = 0.360, *F* (10, 274) = 15.544, *p* < 0.001). After controlling for the sample characteristics in the final model (Model 3), perceived beliefs, practice for change subscale, and emotional transformation subscale were the significant predictors of the likelihood of completing the recommended series of HPV vaccination (*p* < 0.001), with each variable accounting for 7.8%, 8.1%, and 1.1% of the variance respectively.

## 4. Discussion

In the current cross-sectional study, we investigated predictors of Ghanaian adolescents’ intention to accept HPV vaccination. Key predictors for initiating vaccination included age, gender, perceived beliefs, and change in the physical environment. The predictors for completing the recommended doses of HPV vaccination included practice for change and emotional transformation.

Most of the participants in our study reported that they had not heard about HPV nor HPV vaccination. We also found age, gender, and perceived beliefs about the HPV to be significant predictors for initiating vaccination uptake. These finding are consistent with other studies conducted in low middle-income countries. Researchers report a general lack of awareness about HPV and the availability and need for the HPV vaccine [40,41,42,43]. Furthermore, they report an association between the participant’s gender [44] and age [45,46] and the likelihood to initiate HPV vaccination, with females more likely than males to report a willingness to initiate and complete HPV vaccination [44,47]. In our study, perceived beliefs that HPV causes cancer, that HPV vaccination is effective in preventing cancer, and that HPV vaccination is effective in adolescents’ early years were significantly associated with the initiation and completing of HPV vaccination; findings consistent with the previous research [48,49,50,51].

New findings of interest from our study are (1) the effects of the physical environment on HPV vaccination uptake, (2) how practicing for change predicts completion of the HPV vaccination series, and (3) the effects of emotional transformation on willingness to complete the vaccination uptake. Specifically, we found that participants’ ability to make arrangements for payment, having a place to receive the vaccination, and transportation for HPV vaccination were significantly associated with intentions to obtain the vaccine. Other studies identified physical environment and transportation as a barrier to HPV vaccination [52,53]; however, the findings of our study support that change in those physical environment factors (transportation, cost, and place) could increase the likelihood of HPV vaccination initiation.

We also found that practicing for change was a significant predictor of completing the vaccination series. Adolescents practicing for change, such as keeping a journal, increases the likelihood of completing the vaccination series. Similar ideas, such as health care providers sending calendar reminders, have been used to increase health behaviors [54,55,56].

Finally, we found that emotional transformation (i.e., adolescents channeling their emotions, practicing self-motivation, and overcoming self-doubt to getting the recommended doses) significantly influences the adolescent’s willingness to complete the recommended doses of the vaccination. Lloyd et al.’s [57] study among adolescents in London schools found no association between emotion and vaccination acceptance, on the contrary in the Ghanaian adolescent population, we observed that adolescents’ emotional transformation significantly influences their willingness to complete the recommended doses of the vaccination.

### 4.1. Implications

The current study has important implications for health care providers and interventions aimed at promoting the initiation and completion of HPV vaccination among adolescents in Ghana. We observed low knowledge and awareness about HPV despite the fact that there are vaccinations available to prevent the effect of HPV diseases. Therefore, it is critical to design and implement intervention studies to educate Ghanaian adolescents. In other studies, effective interventions have focused on education regarding the etiology and provided epidemiological evidence of HPV, its related diseases, available vaccines, and their effectiveness to prevent those diseases [40,58,59,60].

Our findings underscore the gender differences in the likelihood of accepting HPV vaccination. Girls were more likely to report a willingness to initiate and complete the vaccination series. While it is important to educate females about HPV vaccination, it is equally important to develop intervention studies to address other HPV related cancers. Researchers have found that adolescent boys’ willingness to accept vaccination may be enhanced by knowledge of penile, anal, mouth, and throat cancers and the effectiveness of the vaccination in preventing those types of cancers. That said, this result in gender differences should be interpreted with caution because of the small sample size for the adolescent boys.

Our study findings also highlight the importance of the physical environment. Our data suggest that changing transportation, payment for vaccination bills, and a place for vaccination may influence adolescents’ vaccine intentions. Though these physical environments are modifiable, for most people in low-and middle-income countries, access to transportation to vaccination facilities, and paying for vaccination bills are challenges that would need government support or policy change to allow health insurance coverage to include HPV vaccination [9].

Additionally, our findings underline the need for clinicians and intervention studies to encourage adolescents to practice the habit of keeping records or journals to remind themselves of the next vaccination scheduled visit, practice the strategies to overcome HPV vaccination barriers, and adjust their regular schedules to accommodate vaccination schedule to enhance the chances of adolescents completing of the recommended series. We also observed that emotional transformation is a significant predictor of adolescents’ likelihood of completing the HPV vaccination series. Future intervention is needed to address the adolescents’ emotional transformation by helping them channel their feelings and emotions, teaching self-motivation, and strategies to overcome self-doubt to enhance their probability of completing the HPV vaccination series.

### 4.2. Limitations and Strengths

The current findings should be considered in light of the study’s limitations and strengths. The limitations include the nature of the study design as a cross-sectional study because it can introduce response, recall, and selection biases. Our outcomes are based on self-reported intention to initiate and complete the HPV vaccine series. We were unable to verify vaccination status through vaccine records. The use of a convenience sample limits the generalizability of the study findings to only study participants. The small sample size of 25 adolescent boys is a limitation in the study. A future study should include a larger sample size of boys to better understand the association of sex and HPV vaccination uptake. Finally, the lack of data from parents who are key in decision making for vaccination is a major limitation. However, knowing factors that influence adolescents’ likelihood of accepting the HPV vaccination can help guide a future intervention to facilitate adolescents-parent communication about HPV vaccination [61]. Also, predicting HPV vaccination behavior among adolescents is consistent with several studies [62,63,64]. Notwithstanding, the strengths of the study include the application of a new emerging theory, multi-theory model. Though MTM is a newer theory, the constructs have been proven and tested to have high predictive power. Also, the use of MTM allows us to predict the likelihood of initiation and completion of the recommended doses as those two outcome variables are exclusive but are necessary conditions for determining the efficacy of the vaccination in preventing HPV related cancers. In our study, we focused on adolescents in a low- and middle-income country, Ghana, who have been underrepresented in the HPV vaccination studies. The use of an adequate sample size in our study adds strength to the statistical data analyses. Finally, we identified significant modifiable predictors (perceived beliefs, practicing for change, and emotional transformation) for adolescents’ HPV vaccination initiation and completion.

## 5. Conclusions

In conclusion, the findings underscore the lack of awareness of HPV related cancers among adolescents in Ghana. The likelihood of the HPV vaccination initiation key predictors included perceived beliefs and a change in the physical environment. Perceived beliefs, practice for change, and emotional transformation were key predictors of HPV series completion. Future interventions should address these modifiable factors to increase HPV vaccination uptake among adolescents in Ghana.

## Figures and Tables

**Table 1 ijerph-17-00571-t001:** Demographic characteristics and HPV related knowledge of the participants (*n* = 285).

Variable	Frequency	Percentage
Age	Mean = 15.47	SD * = 1.80
Gender		
Male	25	8.8
Female	260	91.2
Education		
Junior High School	114	40.0
Senior High school	171	60.0
Religion		
Christianity	276	96.8
Muslim	9	3.2
Insurance		
Yes	285	100
No	0	0
Family doctor		
Yes	43	15.1
No	242	84.9
HPV related Knowledge		
Heard about HPV		
Yes	22	7.7
No	263	92.3
Heard about HPV vaccination		
Yes	13	4.6
No	272	95.4

Note: * SD stands for Standard Deviation.

**Table 2 ijerph-17-00571-t002:** Distribution of range, means, and standard deviation of independent and reliability test for the variables and outcome variables (*n* = 285).

Variable		Observed Range				Reliability Tests	
	n	Minimum	Maximum	Mean	SD	Internal Consistency α ^1^	Test–Retest ^2^ (r) ^3^
Perceived beliefs	285	3.00	15.00	10.78	2.81	0.78	0.65
Change in Social Environment	285	3.00	15.00	11.79	2.22	0.88	0.79
Participatory Dialogue	285	−12.00	18.00	2.98	5.30	0.95	0.92
Behavioral Confidence	285	3.00	15.00	10.98	3.11	0.92	0.87
Change in Phy Environ.	285	3.00	15.00	11.79	2.22	0.89	0.81
Emotional Transformation	285	3.00	15.00	11.60	2.82	0.88	0.80
Practicing for change	285	3.00	15.00	11.55	2.86	0.82	0.72
Initiation of HPV vaccination	285	1.00	5.00	3.39	1.49	0.90	0.81
Completion of HPV vaccination	285	1.00	5.00	3.59	1.60	0.96	0.91

Note: ^1^ α denotes Cronbach alpha; ^2^ Test–retest was evaluated using 40 participants; ^3^ Pearson correlation (*r*) was significant at *p* < 0.01.

**Table 3 ijerph-17-00571-t003:** Summary of hierarchical multiple regression analyses for covariates and MTM constructs predicting adolescents’ likelihood of getting first dose of HPV vaccination (*n* = 285).

Variable	Model 1			Model 2			Model 3		
	B	SE B	β	B	SE B	β	B	SE B	β
Gender	−0.59	0.32	−0.11	−0.75	0.31	−0.14 *	−0.38	0.32	−0.07
Age	−0.16	0.07	−0.19 *	−0.15	0.07	−0.18 *	−0.12	0.06	−0.15 *
Education ^NS^	−0.33	0.25	−0.11	−0.23	0.24	−0.08	−0.35	0.23	−0.12
Religion ^NS^	−0.87	0.50	−0.10	−0.73	0.49	−0.09	−0.88	0.46	−0.10
Provider ^NS^	0.45	0.24	0.11	0.45	0.24	0.11	0.15	0.23	0.04
Knowledge	-	-	-	−0.01	0.21	0.00	0.11	0.20	0.03
Perceived beliefs	-	-	-	0.13	0.03	0.25 **	0.07	0.03	0.14 *^,†^
Participatory dialogue ^NS^	-	-	-	-	-	-	−0.03	0.02	−0.10
Behavioral confidence ^NS^	-	-	-	-	-	-	0.04	0.02	0.14
Δ Phy. environment	-	-	-	-	-	-	0.13	0.03	0.27 **^,†^
R^2^	0.069			0.130			0.237		

Note: Δ stands for change; R^2^ = R-squared; F = F-value; NS = non-significant; * *p* < 0.01. ** *p* < 0.001.; Overall model: F (10, 274) = 8.488, *p* < 0.001; ^†^ Change in R^2^ for Perceived beliefs R^2^ = 0.061 and change in physical environment R^2^ = 0.088; Dependent variable is likelihood of getting first dose of HPV vaccination; B = unstandardized coefficient; SE B = standard error of the coefficient; β = standardized coefficient.

**Table 4 ijerph-17-00571-t004:** Summary of hierarchical multiple regression analyses for covariates and MTM constructs predicting adolescents’ likelihood of completing the HPV vaccination series (*n* = 285).

	Model 1			Model 2			Model 3		
Variable	B	SE B	β	B	SE B	β	B	SE B	β
Gender	−1.93	0.32	−0.35 **	−2.13	0.31	−0.38 **	−1.83	0.29	−0.33 **
Age	−0.09	0.07	−0.11	−0.08	0.07	−0.09	−0.09	0.06	−0.10
Education ^NS^	0.16	0.25	0.05	0.27	0.24	0.08	0.09	0.22	0.03
Religion ^NS^	0.35	0.50	0.04	0.51	0.48	0.06	0.43	0.45	0.05
Provider ^NS^	0.38	0.24	0.09	0.38	0.23	0.09	0.10	0.22	0.02
Knowledge ^NS^				−0.24	0.21	−0.06	−0.17	0.19	−0.04
Perceived beliefs	-	-	-	0.16	0.03	0.28 **	0.09	0.03	0.15 **^,†^
Δ Soc. Environment ^NS^	-	-	-	-	-	-	0.02	0.04	0.03
Practice for Δ							0.13	0.03	0.24 *^,†^
Emotional Transformation	-	-	-	-	-	-	0.08	0.04	0.15 **^,†^
R^2^	0.179			0.261			0.362		

Notes: Δ stands for change; R^2^ = R-squared; F = F-value; * *p* < 0.01. ** *p* < 0.001; NS = non-significant; Overall model: F (10, 274) = 15.544, *p* < 0.001; ^†^ Change in R^2^ for Perceived beliefs R^2^ = 0.078, practice for change R^2^ = 0.081, and emotional transformation R^2^ = 0.011; Dependent variable is likelihood of getting first dose of HPV vaccination; B = unstandardized coefficient; SE B = standard error of the coefficient; β = standardized coefficient.

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
