# Peer review of "Multi-Theory Model and Predictors of Likelihood of Accepting the Series of HPV Vaccination: A Cross-Sectional Study among Ghanaian Adolescents"

_ijerph, 2020, doi:10.3390/ijerph17020571_

Round 1

Reviewer 1 Report

The authors have demonstrated that a cross-sectional study of the predictor of the likelihood of accepting HPV vaccination using the MTM for Ghanaian adolescents. After demonstrating projects by GAVI etc., I really understand the problem that the HPV vaccination rate in the country becomes low.

Overall the manuscript is very interesting.  However, some of the information is missing in the text, and the information written below would be critical in the manuscript.

Please show the survey sheet using the quantitative study. The background of Ref 25 and 26 using 44 items Multi Theory Model (MTM) must be different from this study.

How long does it take to complete the participatory dialogue sessions? As the participants are adolescents it would be better to use easier sessions than previous studies for their convenience.

As the participants are adolescents, are there any effects from their parents to improve outcomes? Perhaps, parent occupation might affect the results.

Author Response

Reviewer #1 Comments

The authors have demonstrated that a cross-sectional study of the predictor of the likelihood of accepting HPV vaccination using the MTM for Ghanaian adolescents. After demonstrating projects by GAVI etc., I really understand the problem that the HPV vaccination rate in the country becomes low.

Thank you very much for your comments

Overall the manuscript is very interesting.  However, some of the information is missing in the text, and the information written below would be critical in the manuscript.

Please show the survey sheet using the quantitative study.

We appreciate your comment. We tried to provide details about the questionnaire in the method section but reviewer #3 thinks we have provided too much information and suggested to reduce the method section. To balance the reviewer #1 and reviewer #3 comments, we believe we have provided enough information about the survey. We can provide the survey for the reviewers to look at but at this time we don’t think adding the survey to the manuscript is a good idea.

 The background of Ref 25 and 26 using 44 items Multi Theory Model (MTM) must be different from this study.

Thanks for this comment. In the previous version we stated that we used an “existing instrument” however, the reviewer is correct, we added some new items and also made a few changes to the existing survey. Therefore, we have addressed this concern (see section 2.3, lines 147 – 156, page 4 and now it reads

“We used a modified version of Multi Theory Model (MTM) questionnaire [30]. The reliability of the modified version of the survey was tested using test-retest reliability (correlation coefficient) and internal consistency (Cronbach alpha) methods. To assess the reliability of the tool, the questionnaire was administered twice among 40 adolescents at an interval of two weeks. The correlation coefficient and Cronbach alpha of .70 is considered acceptable [39].  The modified version consisted a 44-item survey assessing demographic characteristics, predictor, and outcome variables. Demographic variables included age, gender, education, religion, and evidence of health insurance. Two knowledge items, whether participants had heard about HPV and HPV vaccinations were included with a yes/no response option.”

How long does it take to complete the participatory dialogue sessions? As the participants are adolescents it would be better to use easier sessions than previous studies for their convenience.

There was no specific session for the participatory dialogue (The name of the construct is "participatory dialogue.") However, we did have a section of the questionnaire that addressed the participatory dialogue (advantages and disadvantages of HPV vaccination). There were ten items for participatory dialogue, and we believe it took less than 6 minutes for the participants to complete those items. In all, it took about 25 to 30 minutes for each adolescent to complete the entire questionnaire depending on the rate at which each participant reads.

As the participants are adolescents, are there any effects from their parents to improve outcomes? Perhaps, parent occupation might affect the results.

Thanks for this concern. Unfortunately, we did not ask any question about parent occupation and therefore could not control it in the study. We have included possible parental influence as a limitation of the study as it could be a possible confounder. We inserted the following sentence in the limitation section (see lines 366 -367, on page 9).

 “Finally, the lack of data from parents who are key in decision making for vaccination is a major limitation”

Reviewer 2 Report

AUTHORS

The study has a strong drawback. Authors address the likelihood for initiation and completion of HPV vaccination focusing on adolescents responses and not their parents or teachers. I have a hard time finding that a 12 year old has the authority to make the decision of being vaccinated or not so I question the validity of the results. For this reason I believe that the approach was wrongly performed and the study did not produce sufficiently relevant data.

A point to check: Variable “insurance” seems to have only 100 responses (Table 1).

Author Response

Our responses are in red font

Reviewer #2 comments

The study has a strong drawback. Authors address the likelihood for initiation and completion of HPV vaccination focusing on adolescents responses and not their parents or teachers. I have a hard time finding that a 12 year old has the authority to make the decision of being vaccinated or not so I question the validity of the results. For this reason I believe that the approach was wrongly performed and the study did not produce sufficiently relevant data.

We appreciate this concern and we know this is a valid concern. We have addressed this concern in the limitation section. We also recognize that this study is important to advance HPV discussions among parents and their adolescents. Please see below the sentences we have inserted in the limitation section (see lines 366 – 370, on page 10).

“Finally, the lack of data from parents who are key in decision making for vaccination is a major limitation. However, knowing factors that influence adolescents’ likelihood of accepting the HPV vaccination can help guide a future intervention to facilitate adolescents-parent communication about HPV vaccination [56]. Also, predicting HPV vaccination behavior among adolescents is consistent with serval studies [57-59].”

A point to check: Variable “insurance” seems to have only 100 responses (Table 1).

The 100 responses was a typo. The participants (n=285, 100%) reported having insurance. This mistake has been corrected. Thank you very much for catching this mistake (see Table 1).

Reviewer 3 Report

Asare et al., in this manuscript present a study entitled "Multi-Theory Model and Predictors of Likelihood of Accepting the Series of HPV vaccination: A cross-sectional study among Ghanaian adolescents". The results here presented are very interesting and significant efforts have been made in this work. Therefore, the results presented in the manuscript may be useful for the scientific community.

Slight Concerns:

It would be helpful if HPV, related to the subject of this study are highlighted, through Virus signs and symptoms, transmission, diagnosis, infection,.. etc.

Line 13: Papilloma Virus United States of America (USA)

Lines 90-91: predict the likelihood of initiation and the likelihood of completion of HPV…. It should be  (predict the likelihood of initiation and completion of HPV….).

Lines 94-95: A target sample size was calculated in the G*Power software, specifying an effect size of .10, an 95 alpha of 0.05 (two-tailed), power of 0.95, and 7 predictors.!! This needs to be checked again.

Lines 103-104: the protocol was reviewed and approved by the Institutional Review Boards (IRBs) of Baylor University and Kwame Nkrumah University of Science and Technology in Ghana. This is repeated in Ethics declarations section (line 352).

Line 109: The participants received a $5.00 gift card (GH₵00 equivalent) for their participation. I think no need to mention this here.

Line 110: the instrument should be the tool.

I think the methodology too long. It is better to be shortened. i.s., The response options, predictor…etc.

Lines 184-187: no need to repeat the values in this manner. You should use Mean ±SD once.

Lines 192. 188, 210, etc.: (see Table 2). I think (Table 2) is enough.

I think tables (3 and 4) need to be rearranged, and their ligands need to be clearer. Authors can combine the two tables (3 and 4) together, especially as the results are very close.

Results of univariate analyses not presented in tables or figures.

Author Response

Our responses are in red font below

Reviewer #3 comments

Asare et al., in this manuscript present a study entitled "Multi-Theory Model and Predictors of Likelihood of Accepting the Series of HPV vaccination: A cross-sectional study among Ghanaian adolescents". The results here presented are very interesting and significant efforts have been made in this work. Therefore, the results presented in the manuscript may be useful for the scientific community.

Thank you very much for these comments

Slight Concerns:

It would be helpful if HPV, related to the subject of this study are highlighted, through Virus signs and symptoms, transmission, diagnosis, infection,.. etc.

We appreciate the fact that the reviewer wants us to give information about the mode of transmission, diagnosis and infection. While these pieces of information are important, we carefully worded our background for the following reason.

We believe since we always associate the HPV vaccination discussions with sexual behavior, it has contributed to the stigma and HPV vaccination hesitancy leading to low vaccination uptake. We have been arguing that we need to reframe the conversations about HPV vaccination as a cancer prevention program instead of talking about individual sexual behavior. To this end, we have developed an intervention using 3r (reframing, reprioritizing and reforming) model to address HPV vaccination stigma and increase HPV vaccination uptake. Our argument is that, we have several vaccinations that we give to our children, for example flu or hepatitis B or C, or malaria vaccinations, without discussing the causes of malaria, flu or hepatitis B before given those vaccinations. In the examples above, we focus on what the vaccination is preventing and not what causes the flu or malaria, or hepatitis B. In the same way, we need to focus on what the HPV vaccination is meant to prevent (e.g. cervical cancer). We wanted to avoid stigma associated with sexuality and HPV vaccination. With that said, we have included the following sentences in the first paragraph of the introduction section (see lines 46 -55, on pages 2).

"HPV is the most common sexually transmitted diseases in the world [1, 2]. HPV infection has a long incubation period with symptoms often occurring years after the initial infection. The latent onset of symptoms is problematic because exposed individuals may be unaware of their disease status yet have the potential to spread the virus to another individual.  Depending on the integrity of the person’s immune system, the virus may be self-eradicated without the person experiencing further health problems [3]. However, persistent infection with oncogenic HPV causes nearly all cervical cancers and many vulvar, vaginal, penile, anal, oral, oropharyngeal, neck, and head cancers [4]. World Health Organization (WHO) reported that there were approximately 528,000 new cases of HPV and 266,000 HPV related deaths worldwide in 2012 [5]."

Line 13: Papilloma Virus United States of America (USA)

Fixed this concern (see line 45 and lines 66-67, on pages  2). It reads

“Human Papilloma Virus”

“United States of America (USA),”

Lines 90-91: predict the likelihood of initiation and the likelihood of completion of HPV…. It should be  (predict the likelihood of initiation and completion of HPV….).

Fixed this concern (see lines 129 – 130, on page 3). It reads

“The purpose of this study was to identify factors that predict the likelihood of initiation and completion of HPV vaccination series in adolescents in Ghana using MTM constructs. “

Lines 94-95: A target sample size was calculated in the G*Power software, specifying an effect size of .10, an 95 alpha of 0.05 (two-tailed), power of 0.95, and 7 predictors.!! This needs to be checked again.

Thanks for this comment. We checked it again and it came up to the same sample size, however we found that it is not a two-tailed so we deleted the two-tailed (see lines 133 – 134, on page 3) and now it reads:

A target sample size was calculated in the G*Power software, specifying an effect size of .10, an alpha of 0.05, power of 0.95, and 7 predictors.”

Lines 103-104: the protocol was reviewed and approved by the Institutional Review Boards (IRBs) of Baylor University and Kwame Nkrumah University of Science and Technology in Ghana. This is repeated in Ethics declarations section (line 352).

We have deleted the whole sentence on line 103 -104 and maintained the sentence in the Ethics declaration section (see line 399 -402).

Line 109: The participants received a $5.00 gift card (GH₵00 equivalent) for their participation. I think no need to mention this here.

We have deleted this sentence

Line 110: the instrument should be the tool.

We have changed “the instrument” to the “the tool” see line 149, on page 4.

I think the methodology too long. It is better to be shortened. i.s., The response options, predictor…etc.

Thank you for this concern. We addressed this concern with the reviewer #1’s comment in mind (see lines 147 -194, on pages 4 & 5). Now it reads

"We used a modified version of Multi Theory Model (MTM) questionnaire [30]. The reliability of the modified version of the survey was tested using test-retest reliability (correlation coefficient) and internal consistency (Cronbach alpha) methods. To assess the reliability of the tool, the questionnaire was administered twice among 40 adolescents at an interval of two weeks. The correlation coefficient and Cronbach alpha of .70 is considered acceptable [39].  The modified version consisted a 44-item survey assessing demographic characteristics, predictor, and outcome variables. Demographic variables included age, gender, education, religion, and evidence of health insurance. Two knowledge items, whether participants had heard about HPV and HPV vaccinations were included with a yes/no response option."

2.4. Predictors

The predictor variables were the constructs of MTM and the perceived beliefs subscale. The MTM constructs consisted of three subscales that predict the initiation of the HPV vaccination: participatory dialogue, behavioral confidence, and physical environment, and three subscales to predict the completion of the HPV vaccination recommended series:  social environment, practice for change, and emotional transformation.

2.4.1. Initiation subscales predictors

The participatory dialogue consists of the perceived benefits of HPV vaccination (5-items) and perceived barriers of HPV vaccination (5-items). The perceived benefits of HPV vaccination items included protection against cervical, head and neck, mouth and throat cancers and genital warts, improve the immune system, and feel better about overall health. The perceived barriers of the HPV vaccination items included side effects, the cost of the vaccine, inconvenience, pain, and possible effects on the reproductive system. To calculate the participatory dialogue score, the disadvantages scores were subtracted from the advantages scores.  This scoring approach is consistent with the recommendation by theory’s author [30], The participatory dialogue score ranged from -25 to 25 where negative scores indicated less likely to participate and positive scores indicated more likely to participate. The behavioral confidence subscale (5-items) included the certainty of getting the HPV vaccination in the future without having insurance, with their present schedule, irrespective of the distance involved to get a vaccination, and vaccine side effects. The changes in physical environment subscale (3-items) included the certainty of arranging for payment, place, and transportation to get the first dose of vaccination. The composite scores were computed for the behavioral confidence and the changes in the physical environment items and the possible scores range were 1 – 25 and 1 – 15 respectively, where higher scores indicated the likelihood of getting the first dose of HPV vaccination.

2.4.2. Completion of recommended dose predictors

The changes in social environment subscale (3-items) includes certainty of getting support from a family member, friends, and health professionals for all the recommended series of HPV vaccination. The practice for change subscale (3-items) included certainty of being able to keep a record to monitor getting the recommended series, overcome barriers to getting recommended doses, and change schedule to be able to get the recommended doses. The emotional transformation subscale (3-items) included certainty related to being able to direct emotions, motivate oneself, and overcome self-doubt to getting the recommended doses. The composite scores were computed for the subscales and the possible score range was 1 – 15 where higher scores indicated the likelihood of completing the recommended doses of the vaccine.

2.4.3. Additional predictor

We also included a perceived belief variable as an additional predictor. The three perceived belief items assessed participants’ perceived belief about whether HPV can cause cancer, HPV vaccination can prevent HPV related cancer, and HPV vaccination was effective when adolescents are vaccinated in their pre-teen years. The possible score was 1 – 15 where higher scores indicated higher beliefs.

Lines 184-187: no need to repeat the values in this manner. You should use Mean ±SD once.

We have addressed this concern and used the mean and SD once (see lines 213 – 223, on page 6)

Lines 192. 188, 210, etc.: (see Table 2). I think (Table 2) is enough.

We have changed to “Table 2” instead of “see Table 2”

I think tables (3 and 4) need to be rearranged, and their ligands need to be clearer. Authors can combine the two tables (3 and 4) together, especially as the results are very close.

Thanks for this suggestion. We attempted to combine the table as suggested by reviewer #3 but we think combining them may be confusing as both are measuring two different outcomes. Table 3 is for initiation and table 4 for completion. We have formatted the tables to fit the page well. We have also clarified the legends.  Now they read:

“Table 3. Summary of hierarchical multiple regression analyses for covariates and MTM constructs predicting adolescents likelihood of getting first dose of HPV vaccination (n=285).”

Table 4. Summary of hierarchical multiple regression analyses for covariates and MTM constructs predicting adolescent likelihood of completing the HPV vaccination series (n=285).

Results of univariate analyses not presented in tables or figures.

We appreciate this concern. We thought creating additional tables will be too many tables and confusing. As such we provided the necessary data for the univariate analyses and creating tables will be redundant, in our opinion.

Reviewer 4 Report

Presented study identified several modifiable predictors of  implementing vaccinations against HPV.  In addition, it applies to the population of low- and middle-income country, with low rates of vaccination and underrepresented in the HPV vaccination studies. The results presented by the authors are consistent with studies from other countries with a similar socioeconomic status. Therefore, it may have important implications for health care providers  and promotion of HPV vaccination among adolescents in Ghana.

I. My general comments:

1. There is no point in the inclusion criteria that the participant should not be vaccinated against HPV (section 2.2), while in the next part of the text the Authors stated: „A sample of 285 unvaccinated adolescents (age 15.47 ± 1.80 years) participated in the study and provided evaluable data …”. Please  update the eligibility criteria.

2. The Authors did not explain the abbreviation „S.D” in Table 1 nor in the main text (where the abbreviation was written in a different form - SD).

3. In the methodology section, there is no information on how the internal correctness was assessed, while the results give the results of the Cronbach’s alpha statistics.

4. Table 3 -  „R” and „F” were not explained; the heading of model 3 is not bolded, why? Same notes apply to the Table 4.

5. There is no information about the statistical significance of the standardized coefficients without asterisks (I guess they are not significant, consider adding the superscript NS - non-signifficant or an explanation in the footer)

6. Line 295 „head and neck, mouth and throat cancers” - seems to be redundant (mouth and throat cancers belong to the head and neck cancers group); consider changing to „… including mouth and throat cancers”.

7. The use of gift cards may be a potential confounder (selection bias), which was not controlled. This should be included in the section on limitations of the study.

8. References must be corrected in accordance with IJERPH requirements.

9. Author Contributions were not described in accordance with CRediT taxonomy.

II. Technical/editorial notes:

1. Abstract is missing;

2. Lines 27-28, 38-40 (and others): inconsistent font sizes;

3. Lines 32-33: „… currently recommends two doses of vaccination for girls aged 9–14 years old …” - word old seems to be redundant;

4. Line 36: „… vaccinated[10] …” - missing space;

5. Please harmonize subheadings (no spaces or dots at the end) - like 3.3 or 3.4

6. It seems to me that the fonts in the tables are not in-line with the IJERPH style; the same applies to the footer of Table 3.

7. I suggest moving table 1 to page 5 (entirely).

8. Consider changing the orientation of the tables (2-4) to vertical (which can ensure that they are placed on one common page).

Author Response

Our responses are in red font below.

Reviewer #4 comments

Presented study identified several modifiable predictors of  implementing vaccinations against HPV.  In addition, it applies to the population of low- and middle-income country, with low rates of vaccination and underrepresented in the HPV vaccination studies. The results presented by the authors are consistent with studies from other countries with a similar socioeconomic status. Therefore, it may have important implications for health care providers  and promotion of HPV vaccination among adolescents in Ghana.

Thank you very much for your comments

My general comments: There is no point in the inclusion criteria that the participant should not be vaccinated against HPV (section 2.2), while in the next part of the text the Authors stated: „A sample of 285 unvaccinated adolescents (age 15.47 ± 1.80 years) participated in the study and provided evaluable data …”. Please update the eligibility criteria.

Updated the inclusion criteria a follows (see lines 139 -140, on page 3).

“To be eligible, adolescents had to be 12–18 years and be able to read or understand English”

The Authors did not explain the abbreviation „S.D” in Table 1 nor in the main text (where the abbreviation was written in a different form - SD).

To be consistent, S.D. in table 1 is changed to SD. Explained abbreviation SD in table 1 as follows

“Note: SD stands for Standard Deviation”

In the methodology section, there is no information on how the internal correctness was assessed, while the results give the results of the Cronbach’s alpha statistics.

This concern has been clearly addressed. Please see subsection 2.3. Measures and Reliability Assessment (see lines 147 -151, on page 4). Now it reads

“We used a modified version of Multi Theory Model (MTM) questionnaire [30]. The reliability of the modified version of the survey was tested using test-retest reliability (correlation coefficient) and internal consistency (Cronbach alpha) methods. To assess the reliability of the tool, the questionnaire was administered twice among 40 adolescents at an interval of two weeks. The correlation coefficient and Cronbach alpha of .70 is considered acceptable [39].  The modified version consisted a 44-item survey assessing demographic characteristics, predictor, and outcome variables. Demographic variables included age, gender, education, religion, and evidence of health insurance. Two knowledge items, whether participants had heard about HPV and HPV vaccinations were included with a yes/no response option. “

Table 3 - „R” and „F” were not explained; the heading of model 3 is not bolded, why? Same notes apply to the Table 4.

We have explained “R” and “F” in the tables (see tables 3 & 4). We have also bolded the heading

There is no information about the statistical significance of the standardized coefficients without asterisks (I guess they are not significant, consider adding the superscript NS - non-signifficant or an explanation in the footer)

We have indicated NS = non-significant in the tables (see tables 3 & 4)

Line 295 „head and neck, mouth and throat cancers” - seems to be redundant (mouth and throat cancers belong to the head and neck cancers group); consider changing to „… including mouth and throat cancers”.

We have changed it to mouth and throat cancer. See line  339 & 340, on page 9.

The use of gift cards may be a potential confounder (selection bias), which was not controlled. This should be included in the section on limitations of the study.

Reviewer # 1 suggested that the gift cards statement was not necessary, so we have deleted that statement from the paper

References must be corrected in accordance with IJERPH requirements.

We have formatted the references to be consistent with IJERPH requirements (see the reference page).

Author Contributions were not described in accordance with CRediT taxonomy.

We have rewritten the authors’ contribution to conform to CRediT taxonomy (see lines 392 – 395). It reads:

 Author Contributions: Conceptualization, M.A., P.A.B., K.A., K.B., and B.A.L.; methodology, M.A., A.B.O., J.R.M., and E.D.P., data collection, M.A., A.K., P.A.B., A.B.O., and M.E.C., data analysis, M.A., P.B., and B.A.L.; writing—original draft preparation, M.A.,  writing—review and editing, all authors reviewed and edited the draft. All authors read and approved the final manuscript.

Technical/editorial notes: Abstract is missing;

Added abstract

Lines 27-28, 38-40 (and others): inconsistent font sizes;

We formatted the inconsistent font sizes

Lines 32-33: „… currently recommends two doses of vaccination for girls aged 9–14 years old …” - word old seems to be redundant;

We have deleted “old”

Line 36: „… vaccinated[10] …” - missing space;

Corrected it.

Please harmonize subheadings (no spaces or dots at the end) - like 3.3 or 3.4

Formatted the subheadings

It seems to me that the fonts in the tables are not in-line with the IJERPH style; the same applies to the footer of Table 3.

Formatted the table and the footer to conform to IJERPH style

I suggest moving table 1 to page 5 (entirely).

Moved table 1 to page 5

Consider changing the orientation of the tables (2-4) to vertical (which can ensure that they are placed on one common page).

Changed the orientation of the tables and they fit a page each.

Round 2

Reviewer 1 Report

I have no specific concern for the current version. 

Reviewer 2 Report

I believe authors have adequately addressed the reviewers concerns. Please correct in line 370 “sevral studies” to “several studies”